# Phenotype selection due to mutational robustness

**Macoto Kikuchi** [ORCID] *

Cybermedia Center, Osaka University, Toyonaka, Japan

* kikuchi.macoto.cmc@osaka-u.ac.jp

## Abstract

The mutation-selection mechanism of Darwinian evolution gives rise not only to adaptation to environmental conditions but also to the enhancement of robustness against mutations. When two or more phenotypes have the same fitness value, the robustness distribution for different phenotypes can vary. Thus, we expect that some phenotypes are favored in evolution and that some are hardly selected because of a selection bias for mutational robustness. In this study, we investigated this selection bias for phenotypes in a model of gene regulatory networks (GRNs) using numerical simulations. The model had one input gene accepting a signal from the outside and one output gene producing a target protein, and the fitness was high if the output for the full signal was much higher than that for no signal. The model exhibited three types of responses to changes in the input signal: monostable, toggle switch, and one-way switch. We regarded these three response types as three distinguishable phenotypes. We constructed a randomly generated set of GRNs using the multicanonical Monte Carlo method originally developed in statistical physics and compared it to the outcomes of evolutionary simulations. One-way switches were strongly suppressed during evolution because of their lack of mutational robustness. By examining one-way switch GRNs in detail, we found that mutationally robust GRNs obtained by evolutionary simulations and non-robust GRNs obtained by McMC have different network structures. While robust GRNs have a common core motif, non-robust GRNs lack this motif. The bistability of non-robust GRNs is considered to be realized cooperatively by many genes, and these cooperative genotypes have been suppressed by evolution.

## Introduction

Living systems in their present form have been created through Darwinian evolution, a process in which a genotype changes so that the phenotype fits the environmental conditions. This process consists of the repetition of a mutation in the genotype and a selection of individuals. This mutation-selection mechanism of evolution not only drives environmental adaptation but also causes a selection bias in evolution. Mutational robustness is one of the best-known biases [1–3].

Mutational robustness is a characteristic trait of living systems that allows them to maintain their functionality despite gene mutations. Comprehensive gene knockout experiments or

**Data Availability Statement:** The source codes and data are found in zenodo as DOI:10.5281/zenodo.10796337.

**Funding:** JSPS KAKENHI Grant Number 23K03261.

**Competing interests:** The authors have declared that no competing interests exist.

rewiring of gene regulatory networks can be used to identify and confirm the robustness of living systems, especially in microorganisms [4–7]. However, the development of robustness through evolution is difficult to investigate experimentally. Therefore, theoretical studies and computational experiments provide important information. A theory based on population dynamics showed that mutational robustness evolves in neutral evolution [8]. To emphasize the fact that this selection bias is different from one determined directly by fitness, this bias is called "second-order" [2]. Second-order selection is considered to enhance evolvability in cases of non-neutral evolution because mutationally robust genotypes have a higher possibility of having offspring with high fitness. The word "evolvability" has been used in many ways [9]; here we used it in the manner of Wagner and Altenberg [10], namely, the ability to make emergence of new traits or new phenotypes.

We note here the usage of the word "fitness." In population genetics, fitness is defined based on the number of offspring, and the effects of all selection biases are taken into account. However, in this study, we defined fitness as a value calculated from a specific fitness function and considered selection biases separately. This definition has been widely used in evolutionary simulations.

Typically, even if more than one phenotype has similar fitness values, the robustness distributions of the corresponding genotypes should differ. Thus, we expect that the second-order selection will affect the appearance probabilities of these phenotypes; phenotypes with relatively robust genotypes will be favored in evolution, whereas phenotypes with relatively fragile genotypes will be suppressed. We expect this phenotype selection, which originates from the difference in the mutational robustness of genotypes, to occur in general situations during evolution. In this study, we considered a model of gene regulatory networks (GRNs) as an example and investigated the phenotype selection induced by the selection bias of mutational robustness using computational methods.

In a previous paper, we showed for a GRN model that mutational robustness is enhanced by evolution, and the appearance of bistability is delayed [11]. These two phenomena are related to each other; our analysis suggested that the reason for the delay in the appearance of bistability was attributed to the fact that the bistable GRNs were less mutationally robust than the monostable GRNs. If we regard monostability and bistability as distinguishable phenotypes, this result may be interpreted as phenotype selection owing to differences in mutational robustness. However, these phenotypes were not yet clearly defined. Thus, in the present study, we extended the previous study to further explore phenotype selection owing to mutational robustness by adopting a different definition of the fitness function, which enabled us to define phenotypes in a clearer manner. Moreover, we conducted stochastic evolutionary simulations that exhibited steady-state evolution, by which we could observe the selection bias of evolution more prominently.

We employed a new methodology proposed in previous studies [11, 12]. Although evolutionary simulation (ES) is the standard method to study evolution numerically, ES alone is not sufficient to discuss selection biases in evolution because the outcomes of ESs already involve the effects of selection biases. To explore the characteristic properties of evolution, we need a reference for comparison with ES. For GRNs, an appropriate reference system is a randomly generated set of GRNs. If there is no selection bias, the properties of the evolutionarily obtained GRNs should coincide with those of the GRNs selected from this set with the same fitness. Ciliberti *et al.* conducted random sampling of GRNs to investigate the structure of the neutral space [13, 14]. However, simple random sampling is not suitable for this study because GRNs with high fitness are rare. Therefore, a rare-event sampling method was required. The author and coworkers proposed to use the multicanonical Monte Carlo (McMC) method for this purpose [11, 12, 15].

McMC was originally proposed for investigating the phase transitions of spin systems [16, 17]. In these cases, McMC enables us to sample the energy evenly over its entire range and is thus particularly effective for studying the first-order phase transitions. Subsequently, it was recognized that McMC can also be used for the rare-event sampling of non-physical systems by considering any quantity as energy [18]. One example involves counting the number of large magic squares [19]. Saito and the author used this method to investigate biological networks [15]. Nagata and the author applied it to GRNs by considering fitness as energy [12]. In this study, McMC enabled us to sample GRNs evenly over the entire range of fitness. A method of comparing the outcomes of ES and the reference set obtained by McMC was proposed by Kaneko and the author [11]. ES and McMC were recently compared to discuss the evolution of genetic codes by Omachi *et al.* [20].

## Model and methods

### Abstract model of GRN

The expression of many genes is regulated by each other in living cells to form complex gene regulatory networks (GRNs). We employed an abstract connectionist-type model of GRNs, in which we expressed a GRN using a directed graph, ignoring the details of gene expression [12, 21–32]. The nodes represent genes, and the edges represent regulatory interactions. Hereafter, we consider one-in-one-out GRNs with one input gene that accepts the input signal from the outside world and one output gene that expresses the target protein. Fig 1 illustrates an example of a small model. In this study, we restricted ourselves to GRNs with the number of nodes $N = 40$ and the number of edges $K = 120$.

For this network model, we assumed the following discrete-time dynamics:

$$x_i(t + 1) = R\left( I\delta_{i,0} + \sum_j J_{ij} x_j(t) \right), \tag{1}$$

$$R(y) = \frac{1}{1 + e^{-\alpha(y-\mu)}}, \tag{2}$$

where $x_i(t)$ represents the expression level of $i$-th gene ($x_i \in [0, 1]$) at $t$-th time step and $J_{ij}$ is the matrix element representing the regulation from $j$-th gene to $i$-th gene. Each element of $J_{ij}$ can take one of three values, 0 and ±1; $J_{ij} = 0$ means that there is no regulation, + 1 is activation,

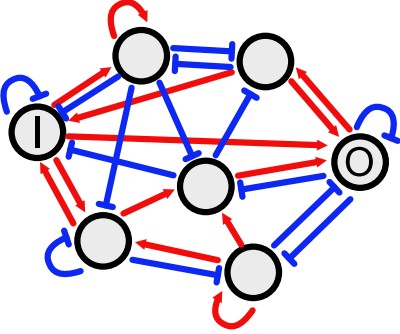

**Fig 1. An example of seven node GRN.** The nodes represent genes, and the edges represent the regulatory interactions. **I** and **O** express the input gene and output genes, respectively. The red lines with arrowheads are activation, and the blue lines with barheads are repression.

and −1 is repression. Self-regulations (the diagonal elements of $J_{ij}$) are permitted. $I$ is the input signal imposed only on the input gene (0-th gene), which is in the range [0, 1]. $R(y)$ is a sigmoidal response function including two parameters $\alpha$ and $\mu$. Throughout this study, we set the values $\alpha = 2.0$ and $\mu = 0.424$, corresponding to the spontaneous expression of each gene being 0.3.

Next, we defined the fitness function different from the one used in the previous studies [11, 12]. For a given GRN, we first set $I = 0$ and set all $x_i$ values at the spontaneous expression level as the initial state. We then ran the dynamics until a steady state was reached. The expression level of the output gene at the steady state for input level $I$ was represented by $x_{out}(I)$. Thus, the first steady state output is $x_{out}(0)$. Note that $x_{out}(I)$ depends on the initial state. Next, we changed the input value to $I = 1$ without changing expression levels. We ran the dynamics again until a new steady state was reached. This output was $x_{out}(1)$. We define the fitness function as the difference between the two output values:

$$f = \max[x_{out}(1) - x_{out}(0), 0]. \tag{3}$$

This fitness function required the output level of the steady state for $I = 0$ to be as low as possible, and that for $I = 1$ to be as high as possible. Using the steady state of $I = 0$ as the initial state for $I = 1$ is important, and the value of $f$ reflects this history. If $x_{out}(1) - x_{out}(0)$ became negative, we considered $f = 0$.

## Multicanonical ensemble Monte Carlo method

Details of the McMC method applied to GRNs have been described in the previous studies [11, 12]. We divided $f \in [0, 1]$ into 100 bins and determined the corresponding weights using the Wang-Landau method [33, 34]. Specifically, we used the entropic sampling method, which is a variant of McMC [35]. From a statistical physics perspective, we assume the law of equal probability and construct a microcanonical ensemble for each fitness bin. Specifically, GRNs in each bin were sampled with the same probability using this method.

To generate a candidate network for the new state, we employed a single-edge rewiring procedure, that is, the cutting of an existing edge selected randomly, followed by the addition of a new edge to a nonconnected node pair that was also selected randomly. The sign of the new edge was selected randomly. With the definition of a Monte Carlo step (MCS) by $K$ Monte Carlo trials, we conducted 16 independent runs of $1.25 \times 10^6$ MCS and 32 independent runs of $2.5 \times 10^6$ MCS resulting in $10^8$ MCS in total. To reduce the correlation between successive samples, we sampled GRNs every 20 MCS. The GRNs obtained for each bin were regarded as randomly sampled.

## Evolutionary simulations

We conducted two types of ES: ESh (where h represents "high fitness") and ESr (where r represents "randomness in selection"). These two methods differ in the selection of genotypes, whereas the generation of new genotypes is the same. For both, the population size was 1000, and the initial population was a randomly generated set of GRNs.

In ESh, 500 GRNs with the highest fitness were preserved in each generation; their copies were produced, and each copy was subjected to a point mutation. We employed the single-edge rewiring procedure described above for point mutation. Because the long run of this method eventually resulted in an extraordinarily high fitness, we stopped the simulation at the 150th generation. Even a short simulation of 150 generations achieved a very high fitness. We selected the GRN with the highest fitness level in the 150th generation and followed its ancestors to obtain a single lineage. We conducted 48000 independent runs and obtained 48000

lineages. The GRNs were classified by fitness into 100 bins, as McMC. This type of simulation provides information about the transient evolution process.

Because it cannot be distinguished by ESh whether the selection bias is intrinsic to evolution or a historical effect of a transient process, it is desirable to study the steady state of evolution. For that purpose, we employed the following simple stochastic selection method in ESr: First, we introduced the "evolutionary temperature" $\beta$. In each generation, each GRN is assigned a value $P = e^{\beta f} \times r$, where $r$ is a uniform random number in [0, 1]. Five hundred GRNs from the highest value of $P$ were preserved, and their copies were subjected to a point mutation similar to that in ESh. We found that the simulation soon reached a steady state. We conducted five independent runs of $10^6$ generations and sampled 500 GRNs from the highest fitness level every 100 generations. After verifying the fitness distribution of the steady state in the preliminary runs, we set $\beta = 0.4$. Because only GRNs in a limited range of fitness appeared by ESr, we show only the data for bins from $f \in [0.85, 0.86)$ to $[0.93, 0.94)$ among the 100 bins in the following section.

## Classification of phenotypes

The GRNs obtained by McMC or ES were classified based on their stability. This model is expected to exhibit the following three distinct stabilities at high fitness [36]: (1) Monostable: When $I$ increases quasi-statically from 0 to 1 and then decreases quasi-statically from 1 to 0, $x_{out}(I)$ follows the same trace in both upward and downward changes. (2) Toggle switch: A Hysteresis appears in some range of $I$ between the upward and downward changes. (3) One-way switch: $x_{out}(0)$ does not return to its initial value, but maintains a high value after a downward change. In dynamical systems, a change in the input signal induces a transition between two fixed points for the bistability to occur. In the case of the toggle switch, saddle-node bifurcations occurs twice in the input range. By contrast, saddle-node bifurcation occurs only once in the allowed range of input in the case of a one-way switch; thus, the transition between the two fixed points is irreversible.

In this study, we did not focus on the biological implications of these stabilities; instead, we regarded these three distinguishable stabilities as three phenotypes for discussing phenotype selection. GRNs obtained by McMC or ES were classified into phenotypes, specifically, stabilities, using the following procedure. For a given GRN, we first set all the expression levels $x_i$ as the spontaneous expression level and set $I = 0$; we ran the dynamics until the steady state was reached. Then, while maintaining the steady-state values of $x_i$'s, we increased $I$ by 0.005 and ran the dynamics again until a new steady state was reached. This procedure was repeated up to $I = 1$. We recorded the trajectory of $x_{out}$ during the upward change. Next, using the steady state of $I = 1$ as the initial state, we repeated a similar procedure inversely from $I = 1$ to 0 to obtain the trajectory of the downward change. If the differences between the two trajectories at all the values of $I$ were less than $10^{-6}$, we regarded the GRN as monostable. If the differences larger than $10^{-6}$ were observed for an intermediate range, we classified the GRN as a toggle switch. If the difference in $x_{out}$ at $I = 0$ at the start and end was greater than $10^{-6}$, the GRN was classified as a one-way switch. Although this classification scheme is not rigorous, it is practical.

## Definition of essential edges

We evaluated the mutational robustness of each GRN by counting the essential edges. An essential edge is defined as the edge at which the fitness drops significantly when cut. The fewer essential edges the GRN has, the more mutationally robust it is. We set the threshold of essentiality to 0.5; that is, when the fitness was less than 0.5 after the cut, the edge was regarded

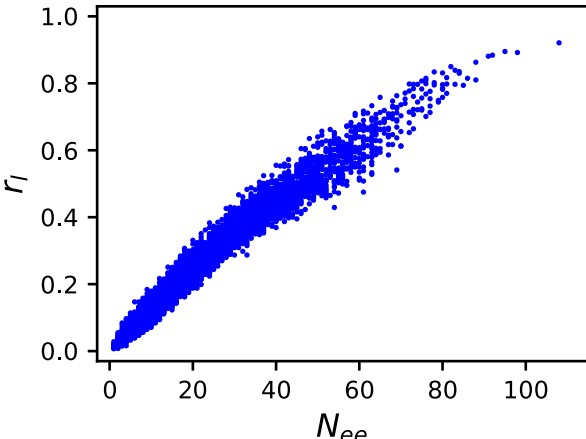

**Fig 2. Correlation of $N_{ee}$ and lethal mutation ratio.** Correlation between the number of essential edges $N_{ee}$ and the ratio $r_l$ of lethal mutations for 10000 GRNs with fitness $f \in [0.9, 0.91]$ obtained by McMC. $r_l$ was estimated from 1000 random mutations for each GRN.

as essential. This definition is valid for GRNs with much larger $f$ than 0.5. We attempted all 120 possible single-edge cuts for each GRN and counted the essential edges to produce an essential-edge distribution.

The mutation procedure conducted in evolutionary simulations is not a single-edge cut but a single-edge rewiring. Then, we took 10000 GRNs with fitness $f \in [0.9, 0.91]$ obtained by McMC, and attempted random single-edge rewiring 1000 times for each GRN to investigate the ratio of lethal mutations, that is, the ratio of mutations that reduced $f$ to below 0.5. The results are shown in Fig 2. The number of essential edges $N_{ee}$ and the ratio of lethal mutations $r_l$ exhibit a strong positive correlation, although it is slightly upward convex. Therefore, $N_{ee}$ is a good indicator of mutational robustness.

## Results

### Genotypic entropy

Fig 3 shows the genotypic entropy obtained using McMC, namely, the logarithm of the number of GRNs, $N(f)$ in each bin. We obtained only the relative entropy, and the entropy for $f \in [0, 0.01)$ was set to zero. The shape of the genotypic entropy is quite similar to that obtained in previous studies, even though the fitness functions are different [11, 12]. We can observe that the majority of GRNs have low fitness, and high-fitness GRNs are rare. This figure shows that GRNs in the entire range of fitness values were successfully sampled.

### Appearance probability of phenotypes

Fig 4(a)–4(c) present examples of the three phenotypes for $f \simeq 0.9$ obtained by McMC. We plotted the change in $x_{out}$ for the quasi-statical change of $I$ for both the upward and downward changes. (a)-(c) correspond to monostable, toggle-switch, and one-way switches, respectively.

Fig 5(a)–5(c) show the appearance probability of the three phenotypes obtained by the three methods. The sum of the probabilities of the three phenotypes for each fitness bin is unity for each simulation method. Because most GRNs in the bin $f \in [0.99, 1]$ obtained by ESh exhibit extraordinarily high fitness, we show GRNs of only $f \in [0.99, 0.991)$ for ESh. The results of McMC represent the ratios of the three phenotypes that should appear in the absence of

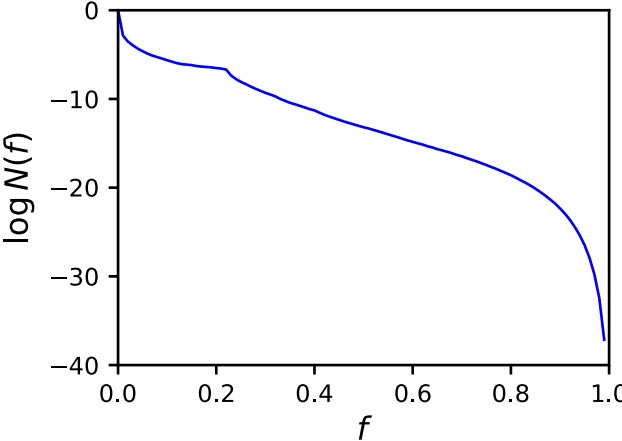

**Fig 3. Genotypic entropy.** Genotypic entropy $\log N(f)$ as a function of fitness $f$ obtained by McMC. $f$ was divided into 100 bins. $N(f)$ is the relative number of GRNs in the corresponding bin, which is normalized so that the entropy is 0 for $f \in [0, 0.01]$.

selection bias. Most GRNs are monostable at low fitness; toggle switches increase as the fitness increases, and finally, one-way switches start to appear. The appearance probabilities of the three phenotypes differed significantly between the three methods. The appearance probabilities of the monostable GRNs obtained by ESh are high up to large values of $f$ compared with those obtained by McMC, and this tendency is more significant for ESr. This suggests that the ratio of monostable GRNs increased with evolution until a steady state was reached. The appearance of the toggle switches was delayed in evolution compared to McMC, and this tendency was more significant for ESr than for ESh. Whereas McMC results show that one-way switches dominated at high fitness in the absence of selection bias, these were suppressed by evolution remarkably, as observed for ESh and ESr. Particularly, in the steady state of evolution (ESr), the appearance of one-way switches was strongly suppressed in the observed range. Thus, the one-way switches were not favored during evolution.

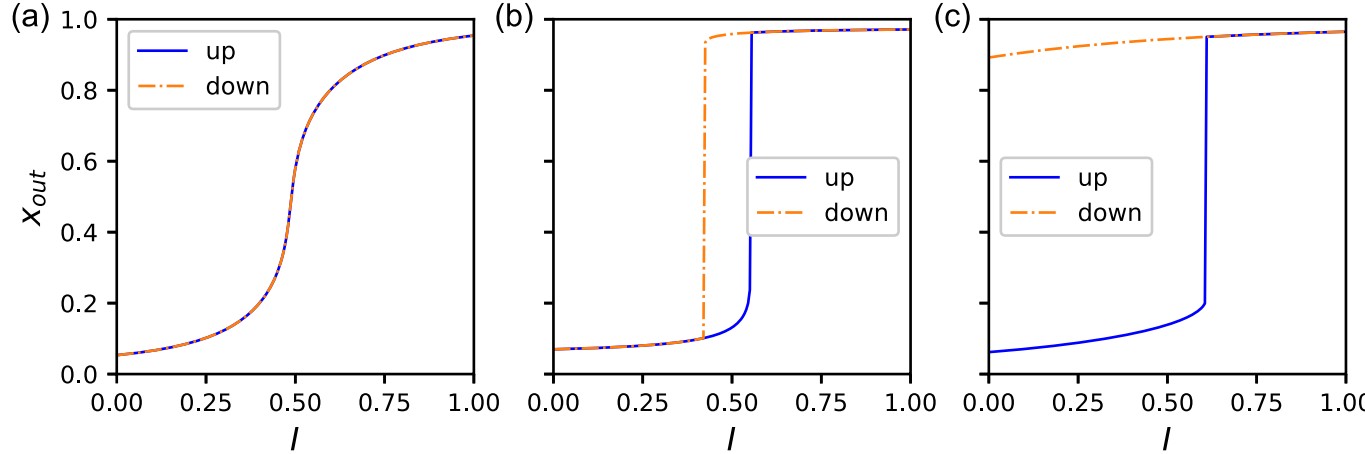

**Fig 4. Examples of three phenotypes.** Three distinct stabilities for $f \simeq 0.9$ obtained by McMC: (a)Monostable (b)Toggle switch (c) One-way switch. The horizontal axes are the input $I$ imposed on the input gene and the vertical axes are the expression of the output gene. The blue solid line indicates the upward change of $I$, and the orange dash-dotted line is the downward change. The protocol for the change of $I$ is described in the Method section.

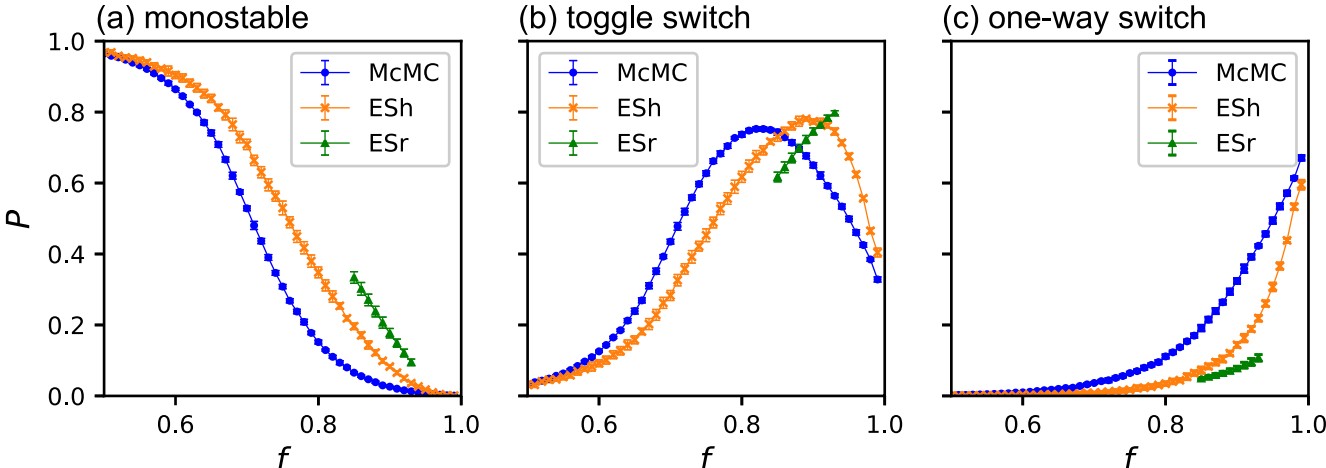

**Fig 5. The appearance probabilities of three phenotypes.** (a)Monostable (b)Toggle switch (c) One-way switch. The circles (blue), the slanted crosses (orange), and the triangles (green) indicate the results obtained by the three methods McMC, ESh, and ESr, respectively. $P$ for the three stabilities for each bin sum as unity for each method. The error bars indicate three times the standard error. For the highest fitness bin of ESh, the result for $f \in [0.99, 0.991]$ is shown.

## Mutational robustness and phenotype selection

To investigate the origin of this selection bias, we counted the essential edges of obtained GRNs. Fig 6(a) shows the probability distributions $P(N_{ee})$ of the number of essential edges $N_{ee}$ for GRNs obtained using McMC in $f \in [0.9, 0.91]$. The distributions were normalized such that the sum of each phenotype was unity. Monostable GRNs had the fewest essential edges. Next, the toggle switches. The number distribution for the one-way switches exhibited a long tail toward a large value of $N_{ee}$. The largest number of essential edges for the one-way switches observed in this study was 109. In other words, the one-way switches include GRNs such that more than 90% of the edges are essential. The same probability distributions obtained by ESr

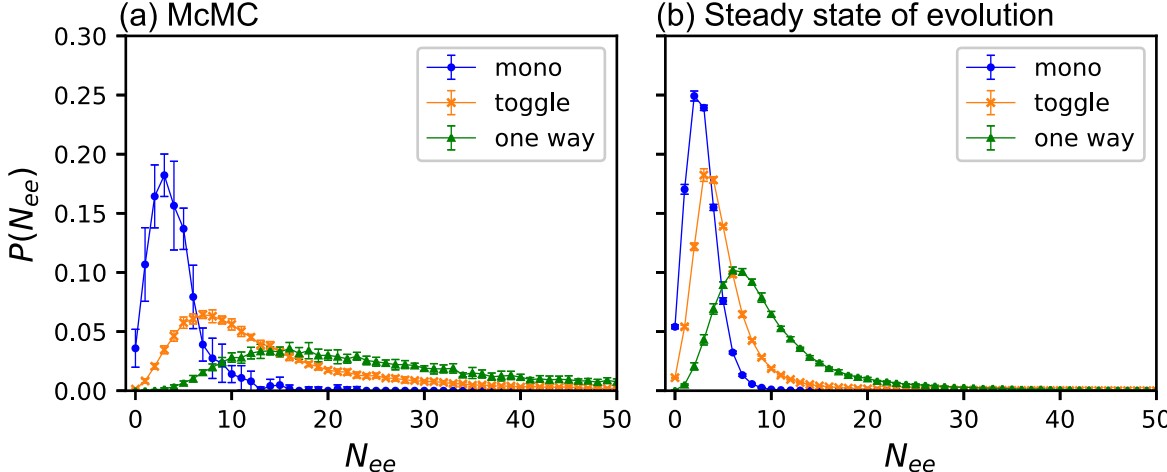

**Fig 6. Essential edge distribution for McMC and ESr.** The probability distributions $P(N_{ee})$ of the number of essential edges $N_{ee}$ for the three phenotypes for $f \in [0.9, 0.91]$: (a) McMC (b) ESr. The circles (blue), the slanted crosses (orange), and the triangles (green) indicate monostable, the toggle switches, and the one-way switches, respectively. $P(N_{ee})$ for each phenotype sum as unity. The error bars indicate three times the standard error.

are plotted in Fig 6(b). This result is considerably different than that obtained by McMC; the distributions are strongly biased toward small values of $N_{ee}$ for all three phenotypes. Therefore, evolution favors mutationally robust GRNs.

We compared the essential edge distributions of the three phenotypes for McMC, ESh, and ESr in Fig 7(a)–7(c). The fitness range is $f \in [0.9, 0.91)$. In contrast to Fig 6, they are normalized such that the sum of all three phenotypes was unity. While the ratio of monostable GRNs increased with evolution, the essential edge distribution did not change significantly compared to McMC. In the case of toggle switches, the distribution shifts toward a small number of essential edges from McMC to ESh and from ESh to ESr. The one-way switches exhibited a remarkable change: GRNs with many essential edges decreased significantly in ESh compared with McMC. The distribution is significantly different for ESr, where only GRNs with a small number of essential edges remain.

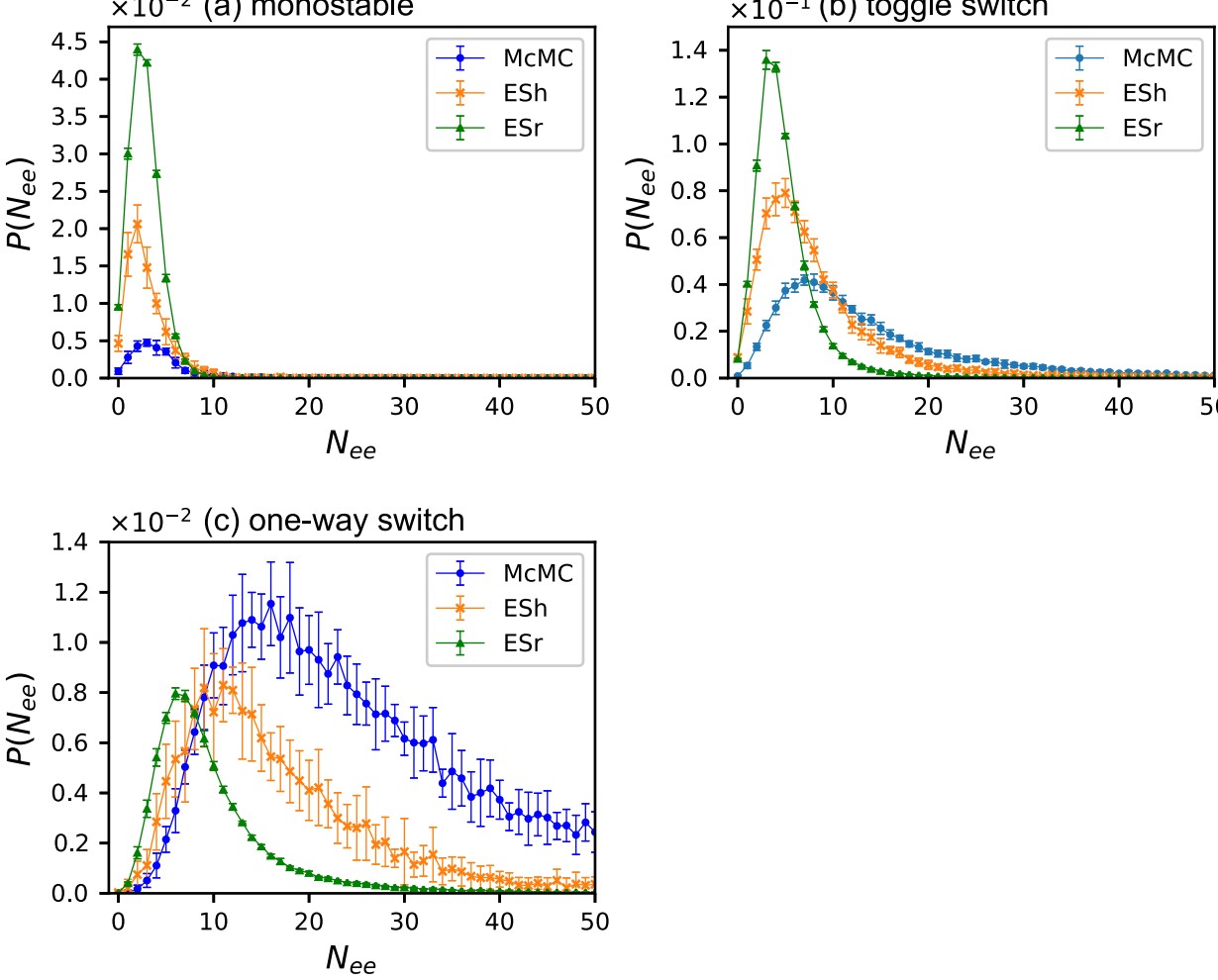

**Fig 7. Essential edge distribution for three phenotypes.** The number distribution of the essential edges for three phenotypes obtained by McMC, ESh, and ESr: (a) monostable (b) the toggle switches and (c) the one-way switches. The circles (blue), the slanted crosses (orange), and the triangles (green) indicate McMC, ESh, and ESr, respectively. In contrast to Fig 6, the sum of probabilities for all three phenotypes for each method is normalized to unity. Note that the scales of the vertical axes for the three figures are different. The error bars indicate three times the standard error.

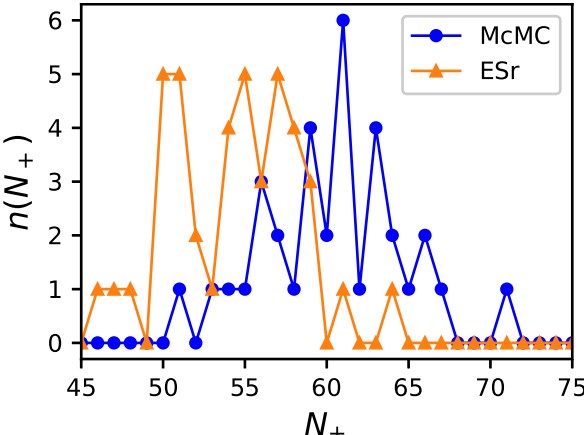

**Fig 8. The number distributions of $J = +1$ edges.** Blue circles is for LNEE and orange triangles is for NOEE. LNEE consists of GRNs of $N_{ee} \geq 90$.

## Network properties

We examined GRNs exhibiting a one-way switch behavior. We compared GRNs with no essential edges obtained by ESr and GRNs with many essential edges obtained by McMC. We call the former set NOEE (no essential edges) and the latter LNEE (a large number of essential edges). NOEE contains 42 GRNs. First, we compared the number of activation edges (*i.e.*, edges with $J = +1$). To ensure good statistics, we took GRNs of $N_{ee} \geq 90$ as LNEE, which contained 34 GRNs. Fig 8 shows the number distributions of $J = +1$ edges for the two sets. The two distributions clearly show a discrepancy: GRNs in NOEE contain more repressive edges on average than those in LNEE. This suggests that the structural characteristics of GRNs in the two sets were different.

Hereafter, we took GRNs of $N_{ee} \geq 96$ (more than 80% of the edges are essential) as LNEE, which consisted of 20 GRNs. Fig 9(a) and 9(b) show the change of $x_{out}$ for changing $I$ from 0 to 1 with the interval of 0.01 for LNEE and NOEE, respectively. While the transition points are

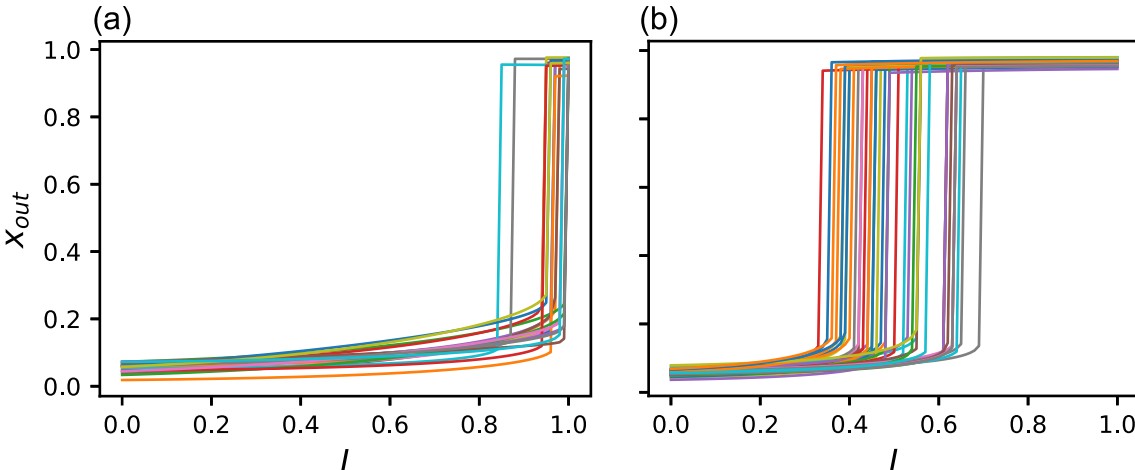

**Fig 9. Response to quasi-static change of I.** Change of $x_{out}$ when $I$ is increased from 0 to 1 with the interval of 0.01. Different colors indicate different samples. (a) LNEE obtained by McMC, which consists of GRNs of $N_{ee} \geq 96$ (b) NOEE obtained by ESr.

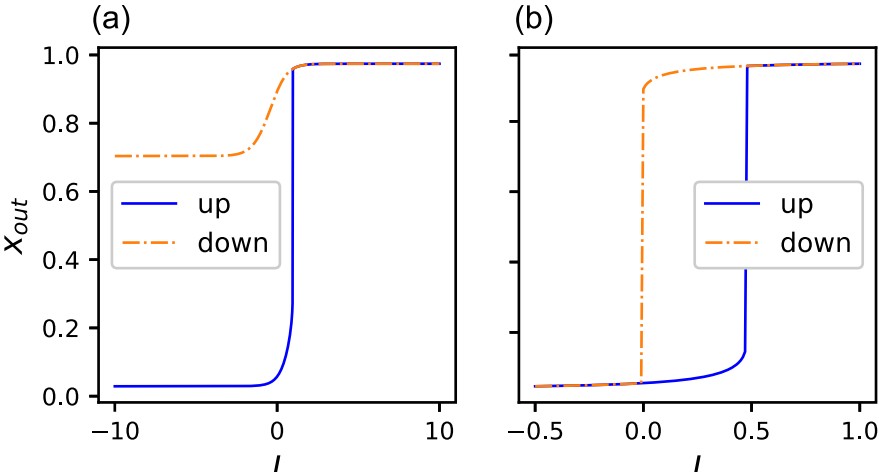

**Fig 10. Response to wider quasi-static change of _I_.** Change of $x_{out}$ when _I_ is changed in wider range upward (blue solid line (and downward (orange dash-dotted line) with the interval of 0.01. (a) The GRN with the maximum number of the essential edges, $N_{ee}$ = 109, in LNEE for $I \in [-10, 10]$. This GRN is an intrinsically one-way switch. (b) A GRN included in NOEE for $I \in [-0.5, 1]$.

near _I_ = 1 for LNEE, they are at middle values of _I_ for NOEE. To investigate the dynamic properties of these GRNs, we extended the input values. Fig 10(a) shows the variations in $x_{out}$ for both upward and downward changes of _I_ in $I \in [-10, 10]$ for GRN with the maximum number of essential edges, $N_{ee}$ = 109, shown in S1 Fig, in LNEE. This GRN has only one saddle-node bifurcation point. This behavior did not change when the input range was further extended to $I \in [-100, 100]$. We identified this as an intrinsically one-way switch (IOWS). The eight GRNS in LNEE are IOWS, whereas the others are toggle switches with a lower bifurcation point at $I < 0$. Fig 10(b) is a similar plot for one GRN, shown in S2 Fig, in NOEE in $I \in [-0.5, 1]$. It exhibits a toggle-switch behavior and becomes a one-way switch in $I \in [0, 1]$ because the lower saddle-node bifurcation is at $I < 0$. All GRNs in NOEE possess the same properties. We may say that a wider bistability range corresponds to a higher cooperativity of the network. Thus, IOWS can be regarded as a limiting case of high cooperativity. These results suggest that the cooperativity tends to be high in GRNs with many essential edges.

We found that 39 GRNs in NOEE contained the motif shown in Fig 11, which we call the ++++ motif, whereas the other three GRNs have three-edge structures lacking one edge of the ++++ motif. Thus, the ++++ motif was the core structure of NOEE. In contrast, no GRN in LNEE possessed the ++++ motif. Only one GRN had a three-edge substructure of the +++

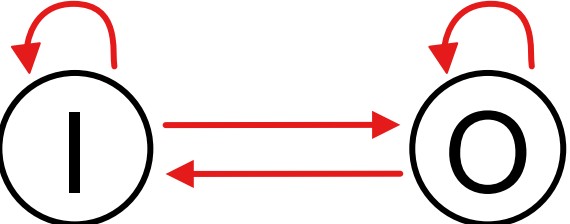

**Fig 11. ++++ motif.** The motif consists of four edges: The auto-activation loops of I and O, and the activation edges of I→O and O→I.

+ motif, while the others had two or fewer edges of the ++++ motif. This suggests that bistability mechanisms are different for LNEE and NOEE. The ++++ motif alone did not exhibit high fitness or bistability; when the input range was extended to $I \in [-10, 10]$, $x_{out}$ changed significantly at $I < 0$, but its change was only $\sim 0.4$. Therefore, for NOEE, edges not contained in the ++++ motif strengthen the cooperativity to realize toggle switches and raise the higher bifurcation point to $I > 0$. We also investigated monostable and toggle switches without essential edges obtained by ESr by sampling 10 GRNs for each case and found that no GRN contained the ++++ motif. This suggests that the ++++ motif is necessary for high cooperativity.

We explored why GRNs in NOEE are mutationally robust. First, it should be noted that cutting the I→O activation edge lowers $f$ of most GRNs to near 0.5, and cutting the auto-activation edge of O lowers $f$ to near 0.7. Thus, these two are important edges, although they are nonessential according to our criteria. We randomly selected one GRN from NOEE, which is shown in S2 Fig. We deleted one randomly selected edge. If the fitness $f > 0.8$ for the resulting GRN, we deleted one more edge. This procedure was repeated as long as $f > 0.8$ was kept. The threshold value of $f$ was greater than that of the former criterion. By repeating this procedure 1000 times, we selected the GRN with the lowest number of edges. As this GRN contained more deletable edges, we repeated the same procedure until no deletable edges remained. Three examples of the reduced networks are shown in Fig 12(a)–12(c). (a) Example of a network in which the ++++ motif remains intact. The network in (b) lacks the auto-activation loop of I; for this case, we can see that a negative-negative two-edge loop constitutes a substitutive path for the auto-activation loop. These are toggle switches. The network in (c) lacks the O→I activation edge; in this case, there is no substitutive path for O→I, and the reduced GRN does not exhibit bistability. We generated seven reduced networks, and found that they belonged to one of these three categories. The structure of substitutive paths was different, and thus, the original GRN has multiple alternative paths.

We also created one reduced network for each of the other 41 GRNs in NOEE and confirmed that 40 reduced networks belonged to one of these three categories. Fig 12 (d) shows the reduced network produced from GRN lacking the auto-activation loop of O. In this case, the auto-activation loop is surrogated by a positive-positive two-edge loop. We also found

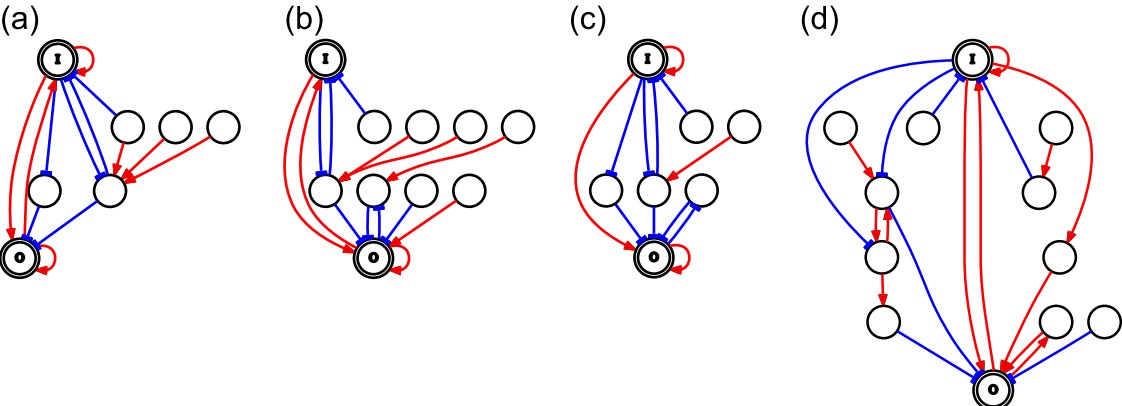

**Fig 12. Reduced networks.** The reduced networks produced from GRNs in NOEE by deleting edges according to the method described in the text. The red lines with arrowhead indicate activation and the blue lines with barhead indicate repression, (a)-(c) are produced from the same GRN shown in S2 Fig. (a) the ++++ motif remains intact (b) auto-activation loop of I is lacking, and a negative-negative two-edge loop serves as its substitutive path. (c) O→I activation edge is lacking, and there is no substitutive path. (d) a reduced network produced from GRN lacking the auto-activation loop of O. A positive-positive two-edge loop surrogates the auto-activation loop.

other surrogating path for the same GRN by other trials. These results suggest that GRNs in NOEE contain the ++++ motif as a core structure and are mutationally robust because of alternative paths. Alternative paths are considered to play also a role in strengthening cooperativity. Under the present robustness criterion, the O→I direct activation is considered necessary.

Next, we investigated the properties of GRN with the maximum number of essential edges in LNEE, as shown in S1 Fig. This network lacks the I auto-activation loop and the O→I activation edge from the ++++ motif. Furthermore, it contains no alternative path surrogating the I auto-activation loop. Therefore, the core structure is different from NOEE. We did not identify a simple motif and considered that this GRN realizes IOWS through the cooperative effect of many edges. When all the nonessential edges were deleted, $f$ remained as high as 0.867. The resulting GRN contained 51 nonessential edges and was more robust than the original network. The six GRNs in LNEE share this property. We then attempted trials of single-edge cut for all 109 essential edges. Although $f$ dropped below 0.5 in all cases, there were two patterns: $x_{out}$ was kept high under a change in $I$ or was kept low. Extending the input range to $I \in [-10, 10]$, we could further categorize them into a few classes: As a result, we found 62 IOWS, 23 toggle switches, and 24 monostable cases. The fact that the majority are IOWS suggests that this GRN is highly cooperative, and that the role of the nonessential edges is to make the transition point within $I \in [0, 1]$.

## Discussion

In summary, the results presented in this paper suggest that one-way switches are rarely selected during evolution because mutationally robust GNRs are favored. This mechanism of phenotype selection was confirmed by comparing the outcomes of the multicanonical ensemble method and the steady state of stochastic evolutionary simulations because the steady state does not depend on evolutionary history. The scenario proposed in this study was natural. Thus, we consider this selection mechanism to be widely valid and not restricted to the case studied here. This phenotype selection can occur whenever more than one phenotype exists for the same fitness value; thus, we expect this phenomenon to occur in broad situations in real living systems. Therefore, mutational robustness may play an essential role in the evolution of certain phenotypes.

Because the second-order selection has a weak effect, the above phenotype selection may be a further weak effect. However, as the present results suggest, this phenomenon may become significant if a steady state of evolution is maintained. Such situations can be experimentally realized. For example, it may be possible to observe the effects on bacteria under constant stress. Multiple peaks in the fitness landscape of *Escherichia coli* have been experimentally observed in the presence of antibiotics [37]. This implies that more than one phenotype has similar fitness. Under such circumstances, phenotype selection due to the present scenario may occur, although this may be difficult to confirm. It should be noted that it matters not the absolute strength of mutational robustness but its relative strength. Thus, this phenotype selection occurs even between mutationally non-robust phenotypes.

For the one-way switches obtained in this study, the network properties differed between the mutationally robust GRNs obtained by evolutionary simulations and non-robust GRNs obtained by McMC. The core structure of the former was a simple motif. Other regulational edges are considered to serve as alternative paths for the motif and play a role in strengthening cooperativity. In contrast, the latter does not possess such a motif. Thus, we believe that the one-way switch of these GRNs is realized due to the complex cooperation of many genes. Therefore, not only a superficial phenotypic trait, but also some network structure, which

represents the genotype in this study, is suppressed by evolution because these genotypes are not mutationally robust.

Next, we consider the biological implications of the evolution of the model in this study. One-way switches of GRNs cause irreversible changes in the states of living cells. They are relevant to processes such as cell maturations and cell differentiation, for example, the maturation of *Xenopus* oocytes [38]. In this case, each fixed point of GRN corresponds to a different cell state. The finding that robust one-way switches were selected may have biological significance in the evolution of irreversible cell state changes. The results of McMC show that most GRNs are monostable at low fitness; then, as fitness increases, the toggle switches dominate, and eventually, most GRNs become one-way switches. We note that irreversibility was not required in the definition of fitness. Thus, irreversible switches can emerge even without explicitly requiring fitness. This suggests that the irreversibility observed in real cells realized by one-way switches evolved from the monostable reversible GRNs.

The present study focused on limited situations, such as a constant environment, a fixed number of genes and edges, and a fixed number of populations in evolutionary simulations. Extending this study to other situations is left for future work.

Finally, the present study demonstrates that McMC is an effective method for investigating the characteristic properties of evolution. McMC clarified the types of phenotypes that may exist if the evolutionary process is not considered. The fact that the outcomes of McMC and evolutionary simulations differ considerably implies that some phenotypes were not reached by evolution. Considering the experimental situation, such phenotypes may be accessible by direct genome engineering rather than by evolutionary experiments. A similar computational method is applicable for analyzing the learning process of neural networks.

## Supporting information

**S1 Fig. An example from 42 one-way switch GRNs having no essential edge obtained by ESr.** The red lines with arrowhead indicate activation and the blue lines with barhead indicate repression.
(PDF)

**S2 Fig. One-way switch GRN with the maximum number of essential edges ($N_{ee}$ = 109) obtained by McMC.** The red lines with arrowhead indicate activation and the blue lines with barhead indicate repression.
(PDF)

## Acknowledgments

The author thank Koichi Fujimoto, Olivier C. Martin, Katsuyoshi Matsushita and Hajime Yoshino for their fruitful discussions and suggestions.

## Author Contributions

**Conceptualization:** Macoto Kikuchi.

**Data curation:** Macoto Kikuchi.

**Formal analysis:** Macoto Kikuchi.

**Funding acquisition:** Macoto Kikuchi.

**Investigation:** Macoto Kikuchi.

**Methodology:** Macoto Kikuchi.

**Project administration:** Macoto Kikuchi.

**Resources:** Macoto Kikuchi.

**Software:** Macoto Kikuchi.

**Supervision:** Macoto Kikuchi.

**Validation:** Macoto Kikuchi.

**Visualization:** Macoto Kikuchi.

**Writing – original draft:** Macoto Kikuchi.

**Writing – review & editing:** Macoto Kikuchi.

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
