## [Decision Letter · Decision Letter 0]

29 May 2024

PONE-D-24-14689Phenotype Selection due to Mutational RobustnessPLOS ONE

Dear Dr. Kikuchi,

Thank you for submitting your manuscript to PLOS ONE. After careful consideration, we feel that it has merit but does not fully meet PLOS ONE’s publication criteria as it currently stands. Therefore, we invite you to submit a revised version of the manuscript that addresses the points raised during the review process.

We look forward to receiving your revised manuscript.

Kind regards,

Steven M. Abel, Ph.D.

Academic Editor

PLOS ONE

Journal Requirements:

"JSPS KAKENHI Grant Number 23K03261"

4. Please expand the acronym “JSPS” (as indicated in your financial disclosure) so that it states the name of your funders in full.

"The author thank Koichi Fujimoto, Olivier C. Martin, Katsuyoshi Matsushita and Hajime Yoshino for their fruitful discussions and suggestions. This work was supported by JSPS KAKENHI Grant Number 23K03261. "

"JSPS KAKENHI Grant Number 23K03261"

Reviewers' comments:

Reviewer's Responses to Questions

**Comments to the Author**

1. Is the manuscript technically sound, and do the data support the conclusions?

Reviewer #1: Yes

2. Has the statistical analysis been performed appropriately and rigorously? 

Reviewer #1: Yes

3. Have the authors made all data underlying the findings in their manuscript fully available?

Reviewer #1: Yes

4. Is the manuscript presented in an intelligible fashion and written in standard English?

Reviewer #1: Yes

5. Review Comments to the Author

Reviewer #1: The author presents a study of gene regulatory networks (GRNs), where the presence of three kinds of dynamical systems behavior are compared between an unbiased sampling of the genotype space (using a Monte Carlo approach) and two kinds of evolutionary simulations. The main finding is that evolution strongly selects against one-way behavior, whilst mono and toggle-switch behavior appear to be favored. The author argues that one-way behavior is less robust than the other two behaviors, in the sense that one-way tends to require more essential edges.

I applaud the massive amount of simulations done to establish a set of ~50k lineages to analyze, and the innovative application of McMC sampling to generate unbiased probability distributions of the three kinds of dynamical systems behavior. The results are not unanticipated, as it is known that (long-term) evolution in a constant environment favors the evolution of robustness.

Overall, the study has been well done and is a nice contribution to the (in silico) evolutionary biology community. In my opinion, it is worthy of publication in PLoS ONE after revisions. Please find below my comments.

Comments:

- citations are put after the sentence; they should be within, i.e. before the end-of-sentence mark.

- please consider having the manuscript checked by a native English person. Especially the use of articles (the) is problematic (too abundant) and hinders a smooth reading experience.

- in the Introduction, it is argued that selection for robustness is almost independent of fitness (line 17). What does the author mean? That there is a trade-off between being fitter and being more robust? I do not remember the precise works that study this relationship, but surely there is such a trade-off and I would expect it to depend on selection pressure.

- in the Introduction, the author states that second order selection enhances evolvability (line 18-19). Since evolvability is such a loosely used term, it would be good if the author defines what exactly they mean in this context, e.g. by choosing one of the three definitions from Pigliucci. It seems, the first definition, evolvability as heritability is the one.

- in the Introduction, the author argues that fitness traditionally is expressed as the number of offspring. I agree, this is surely the case in population genetics. In many simulation studies, however, fitness is a value computed as the author does. So, it is not that special and I think it would be good if the text reflected this more clearly.

- line 106 could be put in Eq. 3 for completeness: f = max(x_out(1) - x_out(0), 0)

- in the Methods, lines 107-119 and lines 120-127 read more like results than Methods. The author should consider moving the text.

- in line 136 it is not clear what happens to the sign of an edge (activation, inhibition) as it is rewired.

- in the Methods, the explanations of ES0 and EST are confusing. The zero in ES0 has nothing to do with the temperature beta of EST, whilst I expected these two to be related. And naming EST explicitly as "stochastic" (line 146) made me think the other one is not... which is not the case, of course, since mutations always introduce some stochasticity. Alternatively, one may argue that "sure mutations are stochastic", but the network is computed in a deterministic manner, so fitness is deterministic and taking the top half of the population is not stochastic either. That is fine with me, but it would be good to make that more explicit, f.i. by stating that here we talk about the selection step only. Some rephrasing is necessary here. Also, at some point I thought EST was a continuation of ES0, but that appears not to be the case either (at least not in operational terms, conceptually yes?). So, do I understand it correctly that ES0 has a classical selection scheme of replacing the bottom-half of the population? And that EST is more like a noisy propertional-fitness selection scheme? I think it would be better either to explain how the two selection schemes are unified (since right now their naming suggests there is a single framework behind) or to choose two really different names, like ES_half and ES_noise, to acknowledge the independent approaches.

- in lines 188 and on, essential edges are defined in a way that evolution never experiences, which may impact the results reported. I mean: removing an edge is normally accompanied by the addition of a new one somewhere else in the network (the definition of a mutation). So to measure mutational robustness by subjecting networks to modifications that are not mutations and that the GRNs never evolved with is inconsistent. It is likely to be rather harmless, but I think this needs to be checked or at least reported as a caveat.

- in the Results, the author does not elaborate on any patterns of network structure that evolution may have come up with. How many edges are activating / inhibiting in evolved and sampled networks? Is the node in- and outdegree of evolved networks very different from the sampled ones? Are there many forcing structures (similar to the AND and OR functions in Boolean networks)? Understanding such structural patterns may give deeper insight into what evolution prefers as a solution to the task of reacting to an input signal.

- Discussion: this study focuses on the "classic study case" of evolution in a constant environment, where a single task needs to be done. There are many other settings that have been left unexplored (e.g. changing environments, multiple tasks, co-evolution, allowing for changes in nr of genes & nr of edges), which makes me rather hesitant at the claim of universality in the Discussion. I know "universality" has a specific meaning in physics, yet I fear a biology or comp sci colleague could easily misunderstand it. I would suggest the author to either not claim it, or better contextualize the claim. Perhaps there are specific experimental results that could be mentioned (E. coli, yeast)? Note that I do not suggest to do more simulations under different kinds of evolutionary scenarios!

- I don't understand why the reported results are third-order. I thought of them as the mechanism (or implementation) that evolution selected. Mutational robustness must manifest in some way. Iow, for me they are second-order.

- it would be good to discuss limitations of the study: besides the above mentioned constant environment, a fixed nr of genes and a fixed total nr of edges is bound to have had an impact on the results presented here.

- the final statement that GAs explore in a biased way the optimization landscape is common knowledge. I would encourage the author to come up with a stronger concluding sentence at the end of their manuscript.

Detailed comments:

- line 2: "developed" is not the term to use, since in biology development is rather different from evolution.

- in figure 1 colors are not explained, especially those of the edges.

- it would be helpful to provide titles in the panels in figures 4 and 6 with "mono", "toggle" and "one-way" and similarly, in figure 5 with "McMC" and "EST". Otherwise, one keeps going back and forth between caption and figure.

- in many panels, errorbars are difficult to see because they overlap with the symbol. Also, too many symbols together make for very fat lines, could the author space them better? Smaller symbols? Less symbols?

- N_e is a famous technical term/symbol in evolutionary biology meaning "effective population size". I advice the author to choose a different one for "nr of essential edges" to avoid confusion. Already N_ee could work, not?

6. PLOS authors have the option to publish the peer review history of their article (what does this mean?). If published, this will include your full peer review and any attached files.

Reviewer #1: No

---

## [Author Response · Author response to Decision Letter 0]

29 Aug 2024

Thank you very much for your careful reading of the manuscript and fruitful comments. I revised the manuscript largely according to your comments. Please read the attached pdf file for details.

---

## [Decision Letter · Decision Letter 1]

12 Sep 2024

Phenotype selection due to mutational robustness

PONE-D-24-14689R1

Dear Dr. Kikuchi,

We’re pleased to inform you that your manuscript has been judged scientifically suitable for publication and will be formally accepted for publication once it meets all outstanding technical requirements.

Kind regards,

Steven M. Abel, Ph.D.

Academic Editor

PLOS ONE

Additional Editor Comments (optional):

Please note that the reviewer has suggested two minor edits, which can be made before publication. 

Reviewers' comments:

Reviewer's Responses to Questions

**Comments to the Author**

1. If the authors have adequately addressed your comments raised in a previous round of review and you feel that this manuscript is now acceptable for publication, you may indicate that here to bypass the “Comments to the Author” section, enter your conflict of interest statement in the “Confidential to Editor” section, and submit your "Accept" recommendation.

Reviewer #1: All comments have been addressed

2. Is the manuscript technically sound, and do the data support the conclusions?

Reviewer #1: Yes

3. Has the statistical analysis been performed appropriately and rigorously? 

Reviewer #1: Yes

4. Have the authors made all data underlying the findings in their manuscript fully available?

Reviewer #1: Yes

5. Is the manuscript presented in an intelligible fashion and written in standard English?

Reviewer #1: Yes

6. Review Comments to the Author

Reviewer #1: I enjoyed reading the revised manuscript and would like to congratulate the author.

Two minor comments:

- An end-of-sentence dot is missing in line 350 (non-colored manuscript version)

- Reference nr 7 first author is "Isalan" not "Isaran"

7. PLOS authors have the option to publish the peer review history of their article (what does this mean?). If published, this will include your full peer review and any attached files.

Reviewer #1: **Yes: **Anton Crombach

---

## [Editor Report · Acceptance letter]

16 Sep 2024

PONE-D-24-14689R1 

PLOS ONE

Dear Dr. Kikuchi, 

I'm pleased to inform you that your manuscript has been deemed suitable for publication in PLOS ONE. Congratulations! Your manuscript is now being handed over to our production team.

Kind regards, 

on behalf of

Dr. Steven M. Abel 

Academic Editor

PLOS ONE